# Proteomic characterisation of the *Chlamydia abortus* outer membrane complex (COMC) using combined rapid monolithic column liquid chromatography and fast MS/MS scanning

**David Longbottom**[ID]\*, **Morag Livingstone, Kevin D. Aitchison, Lisa Imrie**[¤a], **Erin Manson**[¤b], **Nicholas Wheelhouse**[ID][¤c], **Neil F. Inglis**

Moredun Research Institute, Pentlands Science Park, Edinburgh, United Kingdom

¤a Current address: EdinOmics, SynthSys–Centre for Synthetic and Systems Biology, University of Edinburgh, Edinburgh, United Kingdom
¤b Current address: Glasgow Polyomics, University of Glasgow, Wolfson Wohl Cancer Research Centre, Glasgow, United Kingdom
¤c Current address: School of Applied Sciences, Edinburgh Napier University, Edinburgh, United Kingdom
\* david.longbottom@moredun.ac.uk

## Abstract

Data are presented on the identification and partial characterisation of proteins comprising the chlamydial outer membrane complex (COMC) fraction of *Chlamydia abortus* (*C. abortus*)—the aetiological agent of ovine enzootic abortion. Inoculation with the COMC fraction is known to be highly effective in protecting sheep against experimental challenge and its constituent proteins are therefore of interest as potential vaccine candidates. Sodium N-lauroylsarcosine (sarkosyl) insoluble COMC proteins resolved by SDS-PAGE were interrogated by mass spectrometry using combined rapid monolithic column liquid chromatography and fast MS/MS scanning. Downstream database mining of processed tandem MS data revealed the presence of 67 proteins in total, including putative membrane associated proteins (n = 36), such as porins, polymorphic membrane proteins (Pmps), chaperonins and hypothetical membrane proteins, in addition to others (n = 22) that appear more likely to have originated from other subcellular compartments. Electrophoretic mobility data combined with detailed amino acid sequence information derived from secondary fragmentation spectra for 8 Pmps enabled peptides originating from protein cleavage fragments to be mapped to corresponding regions of parent precursor molecules yielding preliminary evidence in support of endogenous post-translational processing of outer membrane proteins in *C. abortus*. The data presented here will facilitate a deeper understanding of the pathogenesis of *C. abortus* infection and represent an important step towards the elucidation of the mechanisms of immunoprotection against *C. abortus* infection and the identification of potential target vaccine candidate antigens.

**Data Availability Statement:** All relevant data are within the manuscript and its Supporting Information files.

**Funding:** The authors gratefully acknowledge financial support from the Scottish Government Rural and Environment Science and Analytical Services (RESAS) division, and by grant BB/E018939/1 from the Biotechnology and Biological Sciences Research Council. We confirm that no funding body had any role in study design, data collection and/or analysis, decision to publish, or preparation of the manuscript.

**Competing interests:** The authors have declared that no competing interests exist.

## Introduction

Chlamydiae are Gram-negative obligate intracellular bacteria that are responsible for a broad range of transmissible diseases affecting both humans and animals [1]. In humans, *C. trachomatis* is the most common cause of venereal infections [2] and trachoma [3], while *C. pneumoniae* is responsible for cases of atypical community-acquired pneumonia [4]. Other chlamydial species cause disease in animals, including *C. psittaci*, which is responsible for psittacosis (aka parrot fever and ornithosis) in psittacine birds and domestic poultry, as well as zoonotic respiratory infections in humans [5]. *Chlamydia abortus* is the aetiological agent of ovine enzootic abortion (OEA), the single most common infectious cause of ovine abortion in the United Kingdom [1] and an important zoonosis posing a potential risk to the health of pregnant women [6–8].

All chlamydial species undergo a unique biphasic developmental cycle and alternate between two distinct morphotypes, represented by elementary bodies (EB) and reticulate bodies (RB), which are adapted to extracellular survival and intracellular replication, respectively. Infection is initiated by the EB attaching to and invading susceptible host cells where it resides within a vacuole known as a chlamydial inclusion, which is non-fusogenic with components of the endocytic pathway [9]. Within the inclusion, the EB converts to the RB which then reproduces through binary fission. After 48–72 hours (depending on chlamydial species) the RB recondenses back into the infectious EB morphotype and are released to invade neighbouring cells [1].

Proteins displayed on the chlamydial cell surface have an important role in host-pathogen interactions and contain epitopes that represent potential diagnostic and vaccine candidate antigen targets. Immunisation with outer membrane proteins extracted from EBs of various chlamydial species as sarkosyl insoluble complexes, known as chlamydial outer membrane complexes or COMCs [10], have been evaluated and shown to have a demonstrable protective capacity in various animal models [11–14]. Specifically, the COMC fraction prepared from *C. abortus* EBs has been shown to protect sheep from experimental challenge with the virulent wild-type strain S26/3 [11]. However, although the protein complement of the *C. trachomatis* COMC fraction is already well documented [15,16], only a few protein components of the *C. abortus* COMC fraction, including the major outer membrane protein (MOMP; a.k.a. Omp1 or OmpA), outer membrane complex protein B (OmcB) and some of the polymorphic membrane proteins (Pmps), have thus far been identified. Accordingly, deeper interrogation of the protein complement of the *C. abortus* COMC fraction is required.

To this end, 2-dimensional gel electrophoresis (2-DGE) combined with MALDI-ToF mass spectrometry (MS) has had success generally in identifying individual proteins in complex biological mixtures [17–19]. However, technical difficulties associated with COMCs, including rate-limiting quantities of sample material and innate target protein hydrophobicity, have constrained progress in this area considerably. Similarly, the application of high resolution on- or off-line nano-flow 2D-liquid chromatography in combination with downstream tandem MS is effectively precluded because of the incompatibility of strongly hydrophobic proteins with the first (ion exchange) dimension. Therefore, this study aimed to circumvent these difficulties by utilising Sawn-Off-Shotgun-Proteomics-Analysis (SOSPA; [20]); a methodology which combines ultra-fast MS/MS scanning with rapid polystyrene-divinylbenzene (PS-DVB) monolithic column liquid chromatography of anionic surfactant-solubilised COMC proteins recovered from entire sample lanes excised from 1D SDS-PAGE gels [21,22]. This approach facilitates the identification and characterisation of intractable hydrophobic proteins, such as those comprising the chlamydial COMC, enabling the downstream development of novel diagnostic assays and immunotherapeutics.

## Materials and methods

### Propagation of *C. abortus* and preparation of the COMC

*Chlamydia abortus* strain S26/3 (available from DSMZ, product number DSM 27085), isolated at Moredun Research Institute in Scotland in 1979 from a vaccinated ewe that aborted [23], was propagated in McCoy cells (obtained from ECACC General Collection in 2005, product number 90010305) in accordance with previously published protocols [24]. Infected cells were harvested using sterile glass beads at 72 h post infection, and EBs (derived from 6 x T225 flasks) were purified by density gradient centrifugation through urografin, as described previously [24]. COMCs were prepared by incubating the purified EBs in 5 mL phosphate buffered saline (PBS), pH 7.4, containing 10 mM EDTA and 2% sarcosyl (sodium N-lauroylsarcosine; Sigma-Aldrich Company Ltd., Dorset, UK) for 1 h at 37°C with occasional mixing and sonication (2 x 5s bursts; Vibra-Cell Processor microtip (Sonics & Materials Inc., Connecticut, US)) to prevent aggregation. The mixture was then centrifuged at 100,000 x g for 60 min to pellet the insoluble material. The pellet was resuspended and further incubated in the same solution containing 10 mM dithiothreitol (DTT; Sigma-Aldrich), under the same conditions. The mixture was then centrifuged as before and washed twice in PBS before finally suspending the insoluble outer membrane complexes in 1 mL PBS and sonicated as above. Resuspended COMC was fractionated by sodium dodecyl sulphate-polyacrylamide gel electrophoresis (SDS-PAGE) and quantified by comparison against BSA standards by densitometry.

### Preparation of COMC proteins for mass spectrometry

Aliquots of COMC material were thawed on ice, mixed with equal volumes of 2x Laemmli loading buffer [25] and incubated at 100°C for 5 min. Solubilised *Chlamydia abortus* COMC proteins (10 μg) were loaded into a single sample well of a discontinuous tris/glycine 10% SDS-PAGE mini-gel and fractionated at 200V (constant voltage) over 45 min using a Mini-Protean™ II Dual Slab Cell (BioRad Laboratories, Hemel Hempstead, UK). Resolved proteins were visualised using SimplyBlue Safe Stain™ (Invitrogen/Thermo Fisher Scientific, Renfrew, UK). The entire stained sample lane was excised, trimmed of excess polyacrylamide and then sliced horizontally from top to bottom to yield a series of 26 equal gel slices of 2.5 mm deep. Each of the resulting gel slices was then subjected to standard in-gel destaining, reduction, alkylation and trypsinolysis procedures [17]. Samples were transferred to HPLC sample vials and stored at 4°C until required for liquid chromatography-electrospray ionisation-tandem mass spectrometry (LC-ESI-MS/MS) analysis.

### LC-ESI-MS/MS

Liquid chromatography was performed using an Ultimate 3000 nano-HPLC system (Dionex/Thermo) comprising a WPS-3000 well-plate micro auto sampler, a FLM-3000 flow manager and column compartment, a UVD-3000 UV detector, an LPG-3600 dual-gradient micropump and an SRD-3600 solvent rack controlled by Chromeleon chromatography software (Dionex/Thermo Fisher Scientific). A micro-pump flow rate of 246μl/min⁻¹ was used in combination with a cap-flow splitter cartridge, affording a $^1/_{82}$ flow split and a final flow rate of 3μl/min⁻¹ through a 5cm x 200μm ID PS-PVD monolithic reversed phase column (Dionex/Thermo) maintained at 50°C. Samples of 4μl were applied to the column by direct injection. Peptides were eluted by the application of a 15min linear gradient from 8–45% solvent B (80% acetonitrile, 0.1% ($^v/_v$) formic acid). LC was interfaced directly with a 3-D high capacity ion trap tandem mass spectrometer (amaZon ETD™, Bruker UK, Coventry, UK) via a low-volume (50μl/

min$^{-1}$ maximum) stainless steel nebuliser (Agilent, cat. no.G1946-20260) and ESI. Parameters for tandem MS analysis were based on those described previously [21].

## Database mining

Deconvoluted MS/MS data in Mascot Generic Format (.mgf) were imported into Protein-Scape™ V3.1 (Bruker UK) proteomics data analysis software for downstream database mining of a cognate *C. abortus* genomic sequence utilising the Mascot™ V2.3 (Matrix Science) search algorithm. The protein content of each individual gel slice was established using the "Protein Search" feature of ProteinScape™, whilst separate compilations of the proteins contained in all 26 gel slices for each sample were produced using the "Protein Extractor" feature of the software. Mascot search parameters were set in accordance with published guidelines [26] and to this end, fixed (carbamidomethyl "C") and variable (oxidation "M" and deamidation "N,Q") modifications were selected along with peptide (MS) and secondary fragmentation (MS/MS) tolerance values of 0.5 Da whilst allowing for a single 13C isotope. Molecular weight search (MOWSE) scores attained for individual protein identifications were inspected manually and considered significant only if: (a) two peptides were matched for each protein; and (b) each matched peptide contained an unbroken "*b*" or "*y*" ion series represented by of a minimum of four contiguous amino acid residues.

## Protein prediction tools

Proteins meeting our confidence criteria were analysed using the PSORT-B protein algorithm that predicts the subcellular localisation of Gram-negative bacterial proteins [27] and the SignalP 4.1 server for predicting the presence and localisation of signal peptide cleavage sites for Gram-negative prokaryotes [28]. Location of passenger, mid and autotransporter domains for members of the polymorphic membrane protein (Pmp) family were estimated using the secondary structure prediction Phyre2 web portal [29] and the prediction of domains using Inter-ProScan 5 [30].

## Results and discussion

In this study we combined rapid monolithic capillary-flow chromatography with ultra-fast MS/MS scanning (aka SOSPA) to identify and characterise the proteins present in the *C. abortus* outer membrane subcellular fraction. Laboratory manipulation of strongly hydrophobic proteins such as those present in the outer membrane is challenging however and not uncommonly, the inclusion of chaotropic agents and/or ionic detergents, which are not MS compatible, is necessary to facilitate their solubilisation in aqueous media. Complicating matters further, the use of these agents at the concentrations necessary to solubilise some hydrophobic proteins can be inhibitory to the action of proteolytic enzymes such as trypsin that are used routinely to cleave proteins into peptides for downstream MS/MS analysis. An earlier study [31], demonstrated that shotgun proteomic assessment of *C. trachomatis* using Gel-LC identified significantly more proteins than either MuDPIT or 2-DGE. Akin to Gel-LC, the SOSPA approach taken in this study exploits the presence of SDS (a strong anionic surfactant) in the first dimension i.e. the Laemmli discontinuous electrophoresis buffer system [25], to solubilise hydrophobic proteins and enable their separation in polyacrylamide gels prior to in-gel enzymatic digestion and downstream analysis of the resulting peptides by tandem mass spectrometry. Although highly similar in principle, the primary difference between SOSPA and regular Gel-LC is the substitution of a traditional 75μm internal diameter (ID) C18 bead matrix reversed phase (RP) column for a 200μm ID PS-DVB RP monolithic column. Modified chromatographic parameters include direct injection of samples i.e. no trap or pre-column as used

in Gel-LC, thus eliminating the loss of hydrophilic peptides, increasing the column flow rate by a factor of 10 and the use of short (15min) elution gradients. Together with fast MS/MS scanning and data acquisition rates, these modifications facilitated the identification of the proteins present in 26 individual gel slices (Fig 1) in less than eight hours. Rapid single-dimensional monolithic chromatography means that SOSPA lends itself particularly well to cataloguing the proteins present in samples of reduced biological complexity e.g. subcellular compartments such as membrane proteins, and combines high sample throughput with time-efficiency and cost-effectiveness.

Detailed proteomic assessment of the protective *C. abortus* COMC fraction unambiguously identified 67 individual proteins, each meeting the stipulated protein identification criteria i.e. at least two peptides per protein with each displaying a minimum of four contiguous amino acid residues represented as an unbroken series of either "*b*" or "*y*" ions (Table 1). These comprise putative membrane associated proteins (n = 36), such as porins, Pmps, chaperonins and hypothetical membrane proteins, as well as others that appear more likely to have originated from other subcellular compartments (n = 22), such as those involved in cellular metabolism.

The proteins in Table 1 are listed in order of confidence as reflected by the number of identified peptides as well as taking into consideration percentage sequence coverage values. As would be anticipated, a significant majority of the proteins featuring at the top of Table 1 are putatively membrane associated. Conversely, those thought to have originated from other subcellular compartments generally appear less prominently and rank lower in Table 1. The identification of what are not considered to be membrane associated proteins *per se*, i.e. those of cytoplasmic origin, has been observed in other studies [15]. The presence of some of these extraneous proteins could be attributed to contamination i.e. the incomplete removal of background cellular proteins during the COMC enrichment protocol. That said, proteins such as Tuf and the chaperonin GroEL have been reported to play a role in virulence in *Chlamydia pneumonia* [32], in other Gram-negative bacteria [33] and may therefore be more closely associated with the cell membrane than previously thought [34]. Other proteins represented by a single confidently identified peptide and generally low sequence coverage are listed in S1 Table. These comprise mainly predicted hypothetical proteins, metabolic enzymes and ribosomal proteins.

## Porins

Integral membrane proteins known as porins are functional water-filled channels common to all Gram-negative bacteria that facilitate the transport of low-molecular weight polar molecules across the outer membrane. All native porins are trimeric in quaternary structure with each monomer typically comprising an eight-stranded anti-parallel β-barrel that is embedded in the outer membrane. In common with other porins, it is noteworthy that OmpA appears as a 110kDa trimer in SDS-PAGE gels when the sample is not heated to 100°C. Conversely, heat denaturation sees the protein appear predominantly as the 40kDa monomer. The chlamydial porin MOMP is unique in that it comprises a number of cysteine residues thought to be involved in the maintenance of structural rigidity of the EB outer membrane via disulphide bonding [35]. MOMP constitutes approximately 60% of the outer membrane protein content which is reflected clearly in the protein profile of the COMC fraction as resolved in SDS-PAGE gels (Fig 1). MS analysis revealed that MOMP is distributed throughout the entire gel lane. Given the relative abundance of MOMP, this observation is perhaps not surprising, as electrophoretic smearing/dragging is not uncommon in these situations. However, although detectable in every gel slice, the protein was most abundant at a migratory position corresponding to the predicted molecular masses of the monomeric (~40kDa; 22 unique validated peptides with

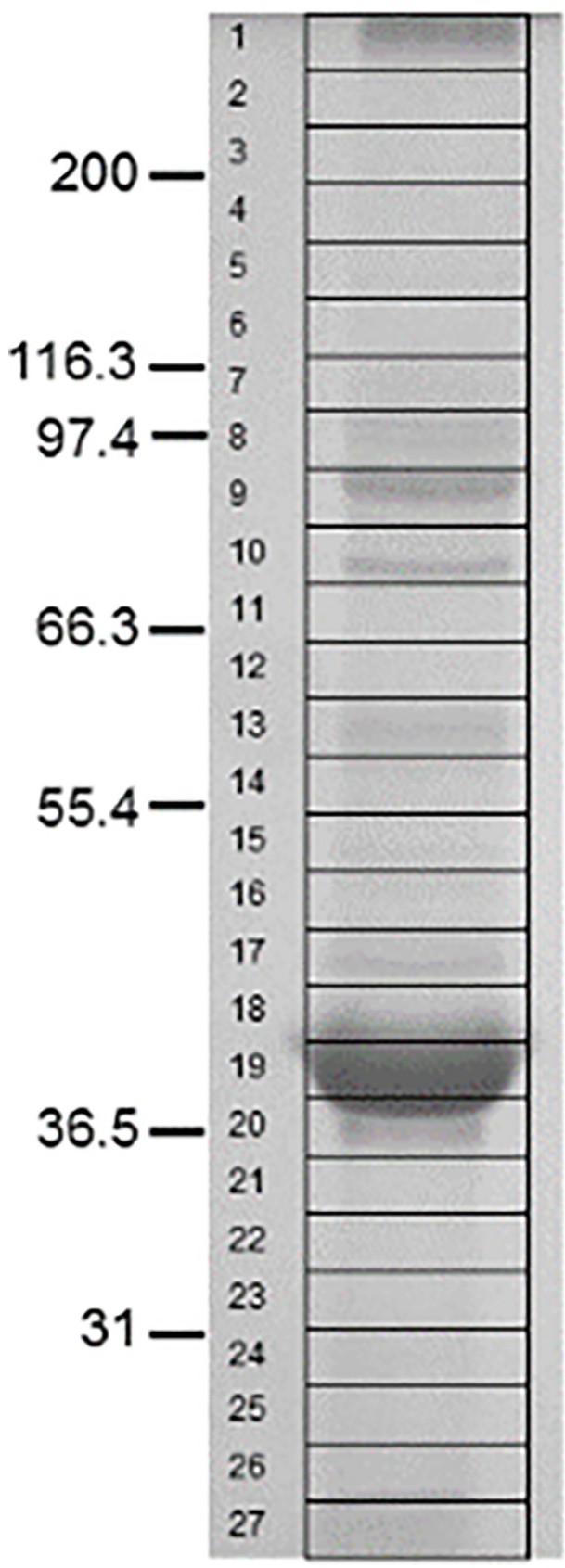

**Fig 1. *C. abortus* S26/3 COMC proteins resolved by SDS-PAGE.** The 26 x 2.5 mm deep gel slices excised are indicated.

77.6% protein coverage) and trimeric (~110kDa; 20 unique validated peptides with 77.6% protein coverage) forms (Table 1).

In common with other Gram-negative bacteria, MOMP plays an important role in bacterial pathogenesis [36], in addition to maintaining structural rigidity and acting as a porin. Antibodies which protect pregnant mice from abortion following *C. abortus* infection [37,38] have been shown to recognise the native trimeric conformation of the MOMP protein. More recently, native *C. trachomatis* MOMP was shown to confer significantly greater protective immunity in a mouse pneumonitis model than a recombinant version of the protein [39]. Although an obvious parallel, a recombinant version of the *C. abortus* MOMP has yet to be trialled as a vaccine candidate in sheep and it therefore remains to be established whether conformational epitopes present only in the natural trimeric structure are prerequisite to immune protection in this species.

Another low abundance porin, known as PorB, which shares weak structural similarity with MOMP was also detected in the COMC fraction (9 unique validated peptides with 31.1% protein coverage; Table 1) [40]. Surface expression of PorB in *C. trachomatis* is constant throughout the cell developmental cycle but a combination of low abundance and relative inefficiency as a general porin has prompted speculation that PorB may function in a substrate-specific manner transporting dicarboxylates for the purpose of fuelling the chlamydial TCA cycle [41]. While the primary structure of MOMP derived from different chlamydial species is known to vary significantly, that of Por1 proteins is more highly conserved [40]. Infection of mice with *C. trachomatis* EBs has been shown to result in a modest serological response to PorB in comparison to MOMP [42] that is possibly due to its low level of expression. However, when combined with MOMP in a multi-subunit vaccine, elevated T-cell and serological responses, as well as an increase in the rate of *C. trachomatis* clearance, was observed than with either PorB or MOMP alone [43].

Demonstrable expression of what was previously considered a hypothetical protein (CAB696) was confirmed by the detection of 9 unique validated peptides. This protein (CTL0626) has also been identified in the *C. trachomatis* COMC [15,16] and conserved domain analysis has predicted it to be a carbohydrate sensitive porin of the OprB family. This protein (CT372) could only be detected in the EB during proteomic analysis of whole chlamydial preparations of *C. trachomatis* strain L2 [31] and it will be of interest to ascertain whether the expression is restricted to this developmental form of the organism in *C. abortus*.

## Type V secretion system

The Type V secretion (or autotransporter) system is a mechanism by which Gram-negative bacteria transport virulence factors to the cell surface or secrete into the extracellular milieu. Autotransporter proteins are expressed as gene products generally possessing an N-terminal signal sequence, as well as two recognisable domains: an effector (or passenger) domain and a C-terminal β-barrel, which facilitates translocation of the effector domain through the outer membrane of the bacterium. *Chlamydia* possess a unique family of Type V proteins known as polymorphic membrane proteins or Pmps [44], which are additionally characterised by a repeat conserved motif in the passenger domain, comprising GG[A/L/V/I][I/L/V/Y] and FXXN, as well as a Pmp-middle domain [45]. The number of Pmps varies in chlamydial species, with 9 present in the human species *C. trachomatis* [46] and 16–21 in other animal species [45,47–49], with the additional Pmps essentially arising from expansion of the Pmp G family.

**Table 1. Proteins from *C. abortus* S26/3 COMC identified with two or more valid peptides.**

| Gene No. | Encoded protein [a] | Predicted molecular mass (kDa) | Predicted cellular localization [b] | Protein coverage (%SC) [c] | No of unique validated peptides | Gel slice(s) [d] | Predicted protein processing [e] |
|---|---|---|---|---|---|---|---|
| CAB200 | Pmp1B | 189,522 | OM & Extra | 31.5 | 34 | 8/9 & 16/17 (25/26) | C |
| CAB181 | OmcB | 59,761 | OM | 70.4 | 26 | 12/13 (1–8, 11, 14–16, 20) | UC |
| CAB268 | Pmp6H | 104,991 | Extra | 40.5 | 23 | 8 (11, 20) | UC C |
| CAB048 | MOMP | 41,884 | OM | 77.6 | 22 | 7/8 & 19 (1–6, 9–18, 20–27) | Trimer & monomer [f] |
| CAB596 | Pmp16G | 90,835 | OM & Extra | 41.3 | 22 | 9/10 (2, 5, 8, 11) | UC |
| CAB041 | SctC | 99,443 | OM | 34.4 | 21 | 8 & 10 (1–7, 9, 11–12) | UC |
| CAB468 | Omp85 | 89,241 | OM | 32.8 | 20 | 10 | UC |
| CAB281 | Pmp13G | 90,695 | OM & Extra | 36.2 | 18 | 9/10 (1, 4, 7–8, 11–14, 16, 19, 22) | UC |
| CAB279 | Pmp12G/17G | 89,824 | OM & Extra | 36.0 | 18 | 9 & 13 (3, 6, 8, 10–12, 14–15, 25) | UC, C |
| CAB265 | Pmp3E | 108,487 | OM & Extra | 26.8 | 17 | 7 (20) | UC C |
| CAB014 | Put. OM protein | 48,237 | OM | 45.3 | 14 | 16/17 | UC |
| CAB668 | tUF | 43,214 | Cyto | 45.4 | 12 | 16 | UC |
| CAB282 | Pmp14G | 98,440 | OM & Extra | 19.2 | 10 | 8 | UC |
| CAB776 | Pmp18D | 163,315 | OM | 16.6 | 10 | 15 & 20 | C |
| CAB269 | Pmp7G | 108,664 | OM & Extra | 14.4 | 10 | 15/16 & 21 | C |
| CAB881 | PorB; OmpB | 37,792 | OM | 31.1 | 9 | 20 | UC |
| CAB696 | Hypo. protein | 50,045 | OM | 28.8 | 9 | 16/17 | UC |
| CAB397 | Put. leucyl aminopeptidase | 54,425 | Cyto | 23.2 | 9 | 14 | UC |
| CAB201 | Pmp2A | 101,808 | OM | 16.3 | 9 | 8 | UC |
| CAB266 | Pmp4E | 104,549 | OM | 16.6 | 8 | 8 | UC |
| CAB661 | rpoB | 140,337 | Cyto | 10.1 | 7 | 6 | UC |
| CAB453 | 30s ribosomal protein S1 (rpsA) | 64,964 | Cyto | 21.5 | 7 | 11 | UC |
| CAB256 | Membrane transport protein | 28,350 | Unknown | 35.4 | 6 | 23 | UC |
| CAB115 | Put. glyceraldehyde 3-phosphate dehydrogenase (gapA) | 36,371 | Cyto | 29.9 | 6 | 20 | UC |
| CAB423 | ABC transporter | 36,428 | CM | 24.9 | 6 | 20/21 | UC |
| CAB615 | Gro-EL 60kDa chaperonin | 58,261 | Cyto | 18.8 | 6 | 12/13 | UC |
| CAB046 | Put. elongation factor | 30,744 | Cyto | 29.4 | 5 | 21 | UC |
| CAB099 | Put. 30S ribosomal protein s3 (rpsC) | 24,512 | Unknown | 26.5 | 5 | 24 | UC |
| CAB267 | Pmp5E | 39,825 | OM & Extra | 24.5 | 5 | 20 | UC |
| CAB283 | Pmp15G | 144,947 | OM & Extra | 6.7 | 5 | 15 | C |
| CAB929 | Hypo. protein | 21,277 | Cyto | 31.0 | 4 | 25 | UC |
| CAB660 | rpoC | 154,965 | Cyto | 5.3 | 4 | 6 | UC |
| CAB886 | Put. helicase | 133,800 | Unknown | 4.4 | 4 | 6 | |
| CAB888 | Clp protease proteolytic subunit | 22,364 | Cyto | 36.1 | 3 | 25 | UC |

*(Continued)*

**Table 1.** (Continued)

| Gene No. | Encoded protein [a] | Predicted molecular mass (kDa) | Predicted cellular localization [b] | Protein coverage (%SC) [c] | No of unique validated peptides | Gel slice(s) [d] | Predicted protein processing [e] |
|---|---|---|---|---|---|---|---|
| CAB839 | elongation factor P (efp) | 21,563 | Unknown | 25.8 | 3 | 25 | UC |
| CAB764 | Put. TMH-family membrane protein | 51,710 | Unknown | 9.8 | 3 | 14 | UC |
| CAB926 | threonyl-tRNA synthetase (thrS) | 73,003 | Cyto | 6.9 | 3 | 11 | UC |
| CAB284 | Put. inner membrane protein | 9,501 | Unknown | 31.9 | 2 | 25 | UC |
| CAB187 | 30S ribosomal protein S7 (rpsG) | 17,744 | Unknown | 25.5 | 2 | 25 | UC |
| CAB817 | Hypo. protein | 17,195 | Cyto | 21.5 | 2 | 25 | UC |
| CAB106 | Put. 50S ribosomal protein l6 (rplF) | 20,093 | Unknown | 19.7 | 2 | 25 | UC |
| CAB083 | Hypo. protein | 27,707 | Cyto | 19.2 | 2 | 23 | UC |
| CAB072 | Put. lipoprotein | 22,153 | Cyto | 18.1 | 2 | 25 | UC |
| CAB395 | Type III secretion chaperone (Slc1) | 18,353 | Cyto | 17.5 | 2 | 25 | UC |
| CAB109 | Put. 50S ribosomal protein l15 (rplO) | 16,093 | Unknown | 16.0 | 2 | 25 | UC |
| CAB531 | ABC transporter, ATP-binding component | 25,484 | CM | 15.5 | 2 | 23 | UC |
| CAB391 | deoxycytidine triphosphate deaminase (dcd) | 21,571 | Unknown | 14.2 | 2 | 25 | UC |
| CAB945 | Put. peptidoglycan-associated protein | 21,873 | OM | 13.1 | 2 | 25 | UC |
| CAB789 | Put. 50S ribosomal protein L25 (rplY) | 20,485 | Cyto | 13.0 | 2 | 25 | UC |
| CAB080 | Put. MIP Lipoprotein | 28,121 | OM | 12.9 | 2 | 21 | UC |
| CAB664 | Put. 50S ribosomal protein L1 (rplA) | 24,862 | Unknown | 12.9 | 2 | 23 | UC |
| CAB393 | Hypothetical protein | 30,388 | Unknown | 12.2 | 2 | 22 | UC |
| CAB818 | 3-oxoacyl-[acyl-carrier-protein] synthase II (fabF) | 44,620 | Cyto | 11.3 | 2 | 16 | UC |
| CAB237 | dnaK | 71,105 | Unknown | 10.2 | 2 | 11 | UC |
| CAB011 | Put. 30s ribosomal protein s4 | 24,051 | Cyto | 10.0 | 2 | 25 | UC |
| CAB341 | Hypo. protein | 52,066 | Unknown | 9.8 | 2 | 14/15 | UC |
| CAB249 | Hypo. Protein | 33,571 | Cyto | 9.7 | 2 | 21 | UC |
| CAB047 | Put. 50S ribosomal protein L2 (rpsB) | 31,234 | Unknown | 8.7 | 2 | 20 | UC |
| CAB452 | nusA | 48,739 | Cyto | 7.4 | 2 | 14 | UC |
| CAB780 | Put. ribonuclease | 59,196 | Cyto | 7.3 | 2 | 12 | UC |
| CAB188 | Put. elongation factor | 76,892 | Cyto | 5.9 | 2 | 10 | UC |
| CAB277 | Pmp10G | 90,458 | OM | 5.8 | 2 | 10 | UC |
| CAB409 | 1-hydroxy-2-methyl-2-(e)-butenyl 4-diphosphate synthase | 67,284 | Cyto | 5.8 | 2 | 12 | UC |
| CAB079 | putative aspartyl-tRNA synthetase (aspS) | 66,483 | Cyto | 5.8 | 2 | 12 | UC |
| CAB308 | Put. lipoprotein | 80,149 | CM | 4.3 | 2 | 11 | UC |
| CAB821 | Hypo. protein | 64,131 | Cyto | 4.0 | 2 | 12 | UC |

(*Continued*)

**Table 1.** (Continued)

| Gene No. | Encoded protein [a] | Predicted molecular mass (kDa) | Predicted cellular localization [b] | Protein coverage (%SC) [c] | No of unique validated peptides | Gel slice(s) [d] | Predicted protein processing [e] |
|---|---|---|---|---|---|---|---|
| CAB503 | Put. exported transferase | 33,494 | Unknown | 2.0 | 2 | 16 | C |

[a] Put., putative; Hypo., hypothetical.

[b] Cyto, cytoplasmic; OM, outer membrane; Extra, extracellular; Unknown, indicates no predicted location.

[c] Protein coverage is expressed as a percentage of total sequence coverage (%SC).

[d] The gel slice that the peptides were detected in (*cf* Fig 1). Slices in brackets represent identification of proteins with reduced confidence (due to electrophoretic dragging of large or very abundant proteins and/or cleavage/degradation).

[e] Indicates whether location of peptides in a particular slice is in agreement with the expected location of the uncleaved (UC) mature protein or is suggestive of potential post-translational cleavage (C).

[f] MOMP in its native non-denatured form has a trimeric quaternary structure with a molecular mass of around 110 kDa when run on a non-denaturing SDS-PAGE gel, while the denatured monomer has a molecular mass of around 40 kDa.

Proteomic studies on *C. trachomatis* have demonstrated expression of all 9 of the Pmp proteins encoded in the genome [31]. The *C. abortus* genome encodes 18 *pmp* genes of which 4 (Pmp8G, Pmp9G, Pmp12G and Pmp16G) have been identified as pseudogenes: Pmp12G is identical to Pmp17G apart from a frame-shift in a polynucleotide tract [45]. This study has demonstrated for the first time expression at the protein level of all translatable Pmps in *C. abortus*, with the exception of Pmp11G. However, although transcription of Pmp11G has been observed at the RNA level, the transcript levels were very low [50], therefore the lack of detectable protein is perhaps unsurprising. It is also possible that there may be differences in the expression of this protein, as well as other proteins, in *in vitro* versus *in vivo* grown pathogen. Consistent with our recent study on Pmp protein expression [51], 22 unique validated peptides of Pmp16G were observed (S1 Fig). While Pmp16G was originally identified as a pseudogene following genome sequencing [45], it was actually identified through screening of a λgt11 expression library [52,53]. This discrepancy arises from the presence of a polynucleotide tract (poly'G') present in the middle domain of Pmp16G and several other closely related members of the G protein family (Pmp12G, Pmp13G, Pmp17G) that may allow phase-variation of expression by strand-slippage [45]. A corroborating observation in *C. pneumoniae* prompted the suggestion that this could be a mechanism of generating antigenic variation in chlamydial species [54].

While the majority of Pmps were identified in gel slices that appeared to correspond to the molecular mass of the mature protein (Pmp2A, Pmp4E, Pmp5E, Pmp10G, Pmp13G, Pmp14G, Pmp16G; peptide coverage indicated in S1 Fig), possible protein cleavage events were observed for a number of others, including Pmp18D (Table 1), for which evidence of post-translational processing has previously been documented [55–58]. In this study, Pmp18D fragments were observed in two gel slices with peptide coverage corresponding to the C-terminal region (Table 2; S1 Fig), within either the middle (M) + autotransporter (AT) domains (slice 15; covering predicted amino acids (aa) 1017–1533) or the AT domain (slice 20; predicted aa 1217–1533). The absence of the N-terminal region (passenger domain) of the protein confirms our earlier observation, which demonstrated cleavage of Pmp18D in *C. abortus* and the solubility of the passenger domain after sarkosyl treatment [58]. It has been suggested that EB surface-associated Pmp18D functions as an adhesin mediating interaction with the host cell and promoting chlamydial attachment or entry and triggering early immunostimulatory events [55,57]. Additionally, it has been proposed that the passenger domain is cleaved and secreted

**Table 2. *C. abortus* S26/3 COMC proteins with evidence of cleavage.**

| Gene No. | Encoded protein | Predicted molecular mass (Da) | Predicted cellular localization [a] | Protein coverage (%SC) [b] | No of unique validated peptides | Gel slice(s) [c] | Predicted protein processing [d] | Peptide coverage [e] | Domains [f] |
|---|---|---|---|---|---|---|---|---|---|
| CAB200 | Pmp1B | 189,520 | OM & Extra | 9.0/14.8 | 8/14 | 8/9 | C | 415–1351 | P+M (148.4) |
| | | | | 11.4/8.1 | 13/8 | 16/17 | C | 1375–1744 | M+AT (58.2) |
| | | | | 3.7/0.8 | 4/1 | 25/26 | C | 231–464 | P (129.1) |
| CAB776 | Pmp18D | 163,613 | OM | 15.0 | 9 | 15 | C | 1081–1530 | M+AT (57.2) |
| | | | | 9.1 | 7 | 20 | C | 1217–1530 | AT (36.5) |
| CAB269 | Pmp7G | 108,663 | OM & Extra | 14.4/6.7 | 10/5 | 15/16 | C | 591–970 | M+AT (56.2) |
| | | | | 6.2 | 5 | 21 | C | 707–970 | AT (35.1) |
| CAB265 | Pmp3E | 108,487 | OM & Extra | 25.5/1.4 | 16/1 | 7/8 | UC | 66–981 | P+M+AT (105.7) |
| | | | | 1.4 | 1 | 15 | C | 619–632 | M+AT (59) |
| | | | | 5.1 | 3 | 20 | C | 758–932 | AT (35) |
| CAB266 | Pmp4E | 104,621 | OM | 13.6 | 7 | 8 | UC | 355–708 | P+M+AT (103.7) |
| | | | | 2.9 | 1 | 20 | C | 842–869 | AT (34.2) |
| CAB268 | Pmp6H | 104,991 | Extra | 38.0 | 22 | 8 | UC | 59–911 | P+M+AT (104.9) |
| | | | | 5.2 | 3 | 11 | C | 350–499 | P+M (67.4) |
| | | | | 5.0 | 2 | 20 | C | 662–936 | AT (32.7) |
| CAB283 | Pmp15G | 144,965 | OM & Extra | 6.7 | 5 | 15 | C | 839–1378 | M+AT (58) |
| CAB598 | Pmp17G | 89,824 | OM & Extra | 34.9 | 18 | 9 | UC | 48–839 | P+M+AT (89.8) |
| | | | | 13.7 | 6 | 13 | C | 348–839 | M+AT (53.6) |

[a] OM, outer membrane; Extra, extracellular.

[b] Protein coverage is expressed as a percentage of total sequence coverage (%SC).

[c] The gel slice that the peptides were detected in (*cf* Fig 1).

[d] Indicates whether location of peptides in a particular slice is in agreement with the expected location of the uncleaved (UC) mature protein or is suggestive of potential post-translational cleavage (C).

[e] Amino acid range of the detected and fully validated peptides for a specific protein (*cf* S1 Fig).

[f] Indicates whether validated peptides are located in gel slices that could include the predicted passenger (P), middle (M) and/or autotransporter (AT) domains (with estimated molecular mass in kDa; in brackets) of the pmp proteins.

from the chlamydial inclusion at a mid to late point in the developmental cycle that impacts on host cell transcription, lysis or inhibition of apoptosis [56,57].

However, to our knowledge there have been no reports on the proteolytic cleavage of any other Pmps in any chlamydial species. In this study, we have additionally identified cleavage of Pmps 1B, 3E, 4E, 6H, 7G, 15G and 17G (Tables 1 and 2; S1 Fig). While Pmps 3E, 4E, 6H and 17G could also be identified as full-length proteins, Pmps 1B, 7G and 15G could only be detected as cleavage products (Table 2). For all of these proteins the data presented in Tables 1 and 2 and in S1 Fig shows peptides present through the whole mature protein sequence (Pmps 1B, 3E, 4E, 6H, 16G, 17G), as well as ones where the peptides cover the P domain (Pmp 1B), P + M domains (Pmps 1B, 6H), M + AT domains (Pmps 1B, 3E, 7G, 15G and 17G) or just the AT domain (Pmps 3E, 6H, 7G). For example, Pmp1B peptides were identified in 3 gel slices (Table 2) corresponding to the P domain (slice 25/26; covering predicted amino acids (aa) 22–

1265), P + M domains (slice 8/9; predicted aa 22–1451) and the M + AT domains (slice 16/17; predicted aa 1266–1788). However, it is clear that the putative passenger domain fragment of Pmp1B, which is estimated to be of molecular mass 129.1 kDa, cannot be present in gel slice 25/26 in an uncleaved form. Therefore, the identification of peptides that correspond to the N-terminal region of the passenger domain in this low molecular mass region of the gel implies that this fragment has been processed further. That said, the authors acknowledge that these putative cleavage events are based entirely on observed mass values and must therefore be viewed as hypothetical. Confirmation of actual endogenous post-translational processing of the Pmps will require validation through further experimentation and precisely targeted proteomic analyses.

## Type III secretion system

All chlamydial species sequenced to date have been found to possess genes encoding the structural apparatus of the type III secretion system (T3SS), found in three conserved genomic clusters [59]. In this study, we confidently identified (34.4% sequence coverage; 21 unique validated peptides) the type III secretion protein C homolog (*SctC*; CAB041) of *YscC* from *Yersinia enterocolitica*. This secretion system enables the translocation of bacterial effector or virulence factors into the target host cell and has been described in many bacterial species, including *Yersinia*, *Shigella* and *Salmonella*. In contrast to the structural apparatus, the effector proteins show little or no sequence homology [60], although they do often share common structural features. In chlamydial species a number of putative T3SS effector proteins have been identified, including Inc (inclusion membrane) proteins and TARP (translocated actin recruiting phosphoprotein) [61]. As these proteins are not directly associated with the outer membrane, we would not necessarily expect to find them in the COMC fraction. However, two proteins were confidently identified, CAB764 and CAB395.

CAB764 is a putative TMH-family protein which possesses N-terminal paired hydrophobic domains characteristic of Inc proteins [45]. These Inc proteins play important roles in the formation of the chlamydial inclusion and may contribute to the growth and survival of the pathogen. However, previous studies in *C. trachomatis* whole organism and COMC preparations have failed to identify the presence of any of the Inc proteins, which is thought to be due to their lack of retention and efficient export [3,31]. While the SOSPA technique is not quantitative, the MOWSE and percentage sequence coverage values for CAB764 were relatively low at 137.8 and 9.8%, respectively, with only three unique validated peptides. This is possibly indicative of the protein being a very minor component of the COMC fraction which was, in all likelihood, captured during its transient secretion through the outer membrane.

The translocation of some of the T3SS effector molecules is mediated through interaction with type 3 secretory chaperones (T3SCs). One of the proteins confidently identified in the COMC preparation was CAB395 (*Slc1*)–a T3SC that has been shown to interact with TARP in *C. trachomatis*, and also in a heterologous model where co-expression of *Slc1* was shown to enhance the translocation of the cognate effector TARP by a surrogate *Yersinia* T3SS [61,62]. Interestingly, TARP (CAB167) was also detected but only with 1 confidently identified peptide and a sequence coverage of 16.9% (S1 Table). Again, as for CAB764 above, this low level of identification may reflect the capture of a low abundance protein during its transient secretion through the outer membrane.

## Other membrane associated proteins

The chlamydial cell envelope structure shares similarity with other Gram-negative bacterial species having an outer membrane containing lipopolysaccharide, a periplasm and an inner

membrane. However, *Chlamydia* possesses two unique features, including an apparent lack of, or deficiency in, peptidoglycan biosynthesis and also the presence of membrane proteins rich in cysteine residues, including MOMP, Pmps and two other cysteine-rich proteins (CRPs), OmcA (9 kDa lipoprotein; gene CAB180) and OmcB (60 kDa; gene CAB181), which may form a periplasmic peptidoglycan-like structure through disulphide-bond cross-linking of these proteins [63,64]. The *C. abortus* genome, in common with that of *C. trachomatis*, does not encode an intact peptidoglycan pathway. However, expression of the *murE* gene product that catalyzes the initial stage in peptidoglycan assembly has been demonstrated in the *C. trachomatis* RB [31]. The presence of this gene in *C. abortus* (CAB835), albeit identified with a single peptide hit (S1 Table), raises the possibility of a peptidoglycan–like molecule in this species. Interestingly, CAB945 (Table 1) has been identified as a putative peptidoglycan-associated protein sharing a 72% identity with *C. trachomatis* Pal, a protein which has previously been demonstrated in the *C. trachomatis* COMC [16]. Pal orthologues in Gram-negative bacteria are involved in the interaction of the outer membrane with periplasmic peptidoglycan and possess several peptidoglycan binding residues that are generally conserved across species, including *C. abortus*, thus providing evidence in support of at least limited peptidoglycan production in this species.

The CRPs are known to be present in the sarkosyl-extracted COMC [64,65], although in this study, where we have prepared the COMC fraction in the presence of the reducing agent DTT, we detected only OmcB, suggesting that OmcA is found only in association with the outer membrane through disulphide-bond cross-linking. It is worth noting that the OmcB sequence coverage (70.4%) and the number of unique validated peptides (n = 26) is second only to MOMP (77.6% coverage; 22 peptides). These values are consistent with the predominance of these proteins as COMC components and their potential as candidate vaccine antigens. Indeed, studies in both *C. pneumoniae* and *C. trachomatis* have demonstrated that chlamydial infectivity can be inhibited by incubation with anti-OmcB antisera; findings that are consistent with its localisation on the chlamydial surface and its identification as a glycosaminoglycan-dependent adhesin [66,67]. In addition, the protein is highly immunogenic and in vitro chlamydial growth has been shown to be completely inhibited by an OmcB-specific CD8(+) T cell clone [68].

Another predominant protein identified within the COMC is the chaperonin Omp85 (32.8% sequence coverage; 20 unique validated peptides). Omp85 family members are found in all Gram-negative bacteria, and are thought to facilitate the integration of β-barrel proteins such as porins and autotransporter proteins into the bacterial outer membrane [69]. The protein structure consists of two-domains, a periplasmic domain and a C-terminal β-barrel that is inserted within the outer membrane. While little is known about its potential protective efficacy, Omp85-specific antibodies neutralize chlamydial infection in vitro [70] and specific IgG responses have been detected in Koalas receiving recombinant *C. muridarum* Omp85 as part of a multi-subunit vaccine [71].

CAB014 (45.3% sequence coverage; 14 unique validated peptides) is a 48 kDa putative outer membrane protein that shares homology with the immunoreactive *C. trachomatis* Ctr48 protein [72] that was demonstrated to be present in the *C. trachomatis* COMC [15,16]. Little is known about the protein's function, although there is homology to a species specific 76kDa protein in *C. pneumoniae* that has been hypothesized to play a potential role in cellular infectivity [73].

Other confidently identified proteins include a number of putative membrane and hypothetical proteins (including CAB014, CAB256, and CAB929). Accordingly, confirmation of their presence in the *C. abortus* COMC as expressed proteins provides conclusive evidence of their existence and elevates their status from hypothetical to validated.

## Conclusion

This study is the first to comprehensively investigate the protein constituents of a *C. abortus* COMC fraction that is known to induce protection against *C. abortus* infection. The data presented herein confirm the presence of 67 individual proteins in total and corroborate the currently accepted biological functions of a number of putative membrane associated proteins, including porins, Pmps and chaperonins. In addition, the confident detection of several other proteins, previously deemed to be hypothetical on the basis of genomic sequence analysis, effectively confirms their expression. Several of the proteins identified in this study are recognised as having potential as diagnostic and/or immunoprophylactic antigens. In reaching this end, the SOSPA approach taken here offers several important advantages. These include: (1) rapid and comprehensive evaluation of a sub-cellular compartment consisting mainly of intractable hydrophobic proteins; (2) the in-gel digestion of smaller and more manageable numbers of proteins concentrated in each of a series of individual gel slices; (3) having fewer peptide ions to cover in any single tandem MS analysis; and (4) generating electrophoretic migratory information, relative to standard protein markers, inferring an approximate molecular mass for each of the proteins identified in any given gel slice. This latter point is especially useful where proteins are cleaved as part of a specific endogenous post-translational processing mechanism as it enables peptides originating from any given protein cleavage product to be mapped to a particular region of the parent precursor molecule. Coupled with detailed amino acid sequence data extracted from peptide fragmentation spectra, this information points towards the approximate location of hitherto unknown protein cleavage sites, as was the case for the Pmp family of proteins described in this study, and ultimately, will lead to their full and unambiguous characterisation.

The data presented here represent a significant step towards elucidating the mechanisms behind post-translational processing of *C. abortus* COMC proteins and overall, serve to enhance current understanding of the pathogenesis of this economically important organism.

## Supporting information

**S1 Fig. Peptide coverage for each of the detected polymorphic membrane proteins.** Predicted signal peptides, passenger domains, middle domains and autotransporter domains are highlighted in red, yellow, blue and green, respectively. All unique confidently identified peptides are in red, vertical lines indicate separation of adjacent peptides. Potential cleavage is indicated by gaps and a scissor symbol between domains in the protein figure below each amino acid sequence. Where there are no gaps between domains plus a scissor symbol this indicates the potential for cleavage but this was not supported by the data presented in this study.
(PPTX)

**S1 File. S1_raw_images.pdf.** Original SDS-PAGE gel visualised with SimplyBlue Safestain and image captured on an AlphaImager 2200 (Alpha Innotech). Lanes: 1, Molecular Weight Markers (Mark12 Unstained Protein Standard, Invitrogen); 2, Bovine serum albumin (BSA); 3, COMC (10μg) sample; 4, COMC (20μg) sample. Lane 3 was used to create Fig 1. Lanes 1, 2 and 4 are not included as denoted by "X".
(PDF)

**S1 Table. Proteins from *C. abortus* S26/3 COMC identified with one unique validated peptide.**
(DOCX)

## Acknowledgments

The authors gratefully acknowledge financial support from the Scottish Government Rural and Environment Science and Analytical Services (RESAS) division, and by grant BB/E018939/1 from the Biotechnology and Biological Sciences Research Council. We confirm that no funding body had any role in study design, data collection and/or analysis, decision to publish, or preparation of the manuscript.

## Author Contributions

**Conceptualization:** David Longbottom.

**Data curation:** David Longbottom, Morag Livingstone, Nicholas Wheelhouse, Neil F. Inglis.

**Formal analysis:** David Longbottom, Nicholas Wheelhouse, Neil F. Inglis.

**Funding acquisition:** David Longbottom.

**Investigation:** David Longbottom, Morag Livingstone.

**Methodology:** David Longbottom, Morag Livingstone, Kevin D. Aitchison, Lisa Imrie, Erin Manson, Nicholas Wheelhouse, Neil F. Inglis.

**Project administration:** David Longbottom.

**Resources:** David Longbottom.

**Validation:** David Longbottom, Neil F. Inglis.

**Writing – original draft:** David Longbottom, Morag Livingstone, Nicholas Wheelhouse, Neil F. Inglis.

**Writing – review & editing:** David Longbottom, Morag Livingstone, Neil F. Inglis.

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
