## [Decision Letter · Decision Letter 0]

7 Oct 2019

Proteomic characterisation of the Chlamydia abortus outer membrane complex (COMC) using combined rapid monolithic column liquid chromatography and fast MS/MS scanning

PONE-D-19-19899

Dear Dr. Longbottom,

We are pleased to inform you that your manuscript has been judged scientifically suitable for publication and will be formally accepted for publication once it complies with all outstanding technical requirements.

With kind regards,

Deborah Dean, M.D., M.P.H.

Academic Editor

PLOS ONE

Additional Editor Comments (optional):

Reviewers' comments:

Reviewer's Responses to Questions

**Comments to the Author**

1. Is the manuscript technically sound, and do the data support the conclusions?

Reviewer #1: Yes

2. Has the statistical analysis been performed appropriately and rigorously? 

Reviewer #1: Yes

3. Have the authors made all data underlying the findings in their manuscript fully available?

Reviewer #1: Yes

4. Is the manuscript presented in an intelligible fashion and written in standard English?

Reviewer #1: Yes

5. Review Comments to the Author

Reviewer #1: In this paper, the authors show a detailed description of the protein components of a Chlamydia abortus fraction of the external membrane complex. C. abortus is considered to be one of the most important miscarriage-causing diseases in small ruminants and, as the COMC fraction has been shown to confer protection on sheep, the identification of the components of this fraction is of great interest for the further development of diagnostic assays and the identification of new vaccine candidates.

The authors have solved the technical difficulties associated with the solubilization of highly hydrophobic proteins by using a novel combined approach (SOSPA).

General comment,

The topic is relevant, the methodology is adequate and the results are interesting and novel. The detailed description of each group of proteins, broadens the understanding we have, both of the functions of these proteins, and of the metabolism of the C. abortus.

6. PLOS authors have the option to publish the peer review history of their article (what does this mean?). If published, this will include your full peer review and any attached files.

Reviewer #1: No

---

## [Editor Report · Acceptance letter]

15 Oct 2019

PONE-D-19-19899 

Proteomic characterisation of the *Chlamydia abortus* outer membrane complex (COMC) using combined rapid monolithic column liquid chromatography and fast MS/MS scanning 

Dear Dr. Longbottom:

I am pleased to inform you that your manuscript has been deemed suitable for publication in PLOS ONE. Congratulations! Your manuscript is now with our production department. 

With kind regards,

on behalf of

Professor Deborah Dean 

Academic Editor

PLOS ONE